# A Cross-Source Point Cloud Registration Algorithm Based on Trigonometric Mutation Chaotic Harris Hawk Optimisation for Rockfill Dam Construction

**DOI:** 10.3390/s23104942

**Published:** 2023-05-21

**Authors:** Bingyu Ren, Hao Zhao, Shuyang Han

**Affiliations:** State Key Laboratory of Hydraulic Engineering Simulation and Safety, Tianjin University, Tianjin 300072, China; zhaoh_2020@tju.edu.cn (H.Z.);

**Keywords:** point cloud registration, Harris hawk optimisation, 3D laser scanning, UAV tilt photography

## Abstract

A high-precision three-dimensional (3D) model is the premise and vehicle of digitalising hydraulic engineering. Unmanned aerial vehicle (UAV) tilt photography and 3D laser scanning are widely used for 3D model reconstruction. Affected by the complex production environment, in a traditional 3D reconstruction based on a single surveying and mapping technology, it is difficult to simultaneously balance the rapid acquisition of high-precision 3D information and the accurate acquisition of multi-angle feature texture characteristics. To ensure the comprehensive utilisation of multi-source data, a cross-source point cloud registration method integrating the trigonometric mutation chaotic Harris hawk optimisation (TMCHHO) coarse registration algorithm and the iterative closest point (ICP) fine registration algorithm is proposed. The TMCHHO algorithm generates a piecewise linear chaotic map sequence in the population initialisation stage to improve population diversity. Furthermore, it employs trigonometric mutation to perturb the population in the development stage and thus avoid the problem of falling into local optima. Finally, the proposed method was applied to the Lianghekou project. The accuracy and integrity of the fusion model compared with those of the realistic modelling solutions of a single mapping system improved.

## 1. Introduction

Rockfill dams are among the types of dams mainly used in plateau areas because the materials for their construction are easy to access. Moreover, they can adapt to the environment and have satisfactory seismic resistance [1]. As sensors continue to advance and technologies (such as artificial intelligence, the Internet of Things, and cloud computing) progressively mature [2], rockfill dam construction has also accelerated in terms of digital and intelligent development. High-precision three-dimensional (3D) models are required for visualising construction simulations and engineering topology carriers. These models are the basis and prerequisite of digitalising water conservancy projects and have been applied to activities, such as dam safety inspection [3], project progress display [4], and augmented reality (AR) model construction [5]. The application of 3D models to water conservancy projects aims to establish the mapping relationship between the natural space entity of a rockfill dam and the virtual space of a computer 3D model. Furthermore, the natural engineering entity is simulated according to the physical equation of change trend over time. Finally, the 3D model is dynamically updated according to the actual construction state to provide dynamic decision making and scheduling control instructions for the reasonable and efficient construction of engineering projects. Presently, various sensors have been applied to rockfill dam mapping during construction. They generate numerous point clouds and images containing spatial and texture information, thus providing a database for the fusion modelling of 3D point clouds from multiple sources of rockfill dam data. Therefore, the rapid acquisition, analysis, and processing of the massive cross-source data provided by sensors during dam construction are necessary. Then, establishing a high-precision, high-integrity, and high-fidelity 3D model of the dam is key to enhancing rationality and efficiency in the construction management and control of rockfill dams.

Unmanned aerial vehicle (UAV) tilt photography and 3D laser scanning are new geodetic technologies developed in recent years, which have overturned traditional manual modelling methods based on computer graphics and empirical knowledge [6]. Their modelling efficiency and fidelity have considerably improved. Using multiple sensors, UAV tilt photography captures images from one vertical and four tilt angles [7,8]. Textures are automatically added using office operations to build a 3D realistic model matching human eye vision [9]. In comparison, 3D laser scanning is a noncontact measurement technology. It rapidly acquires massive point clouds with irregular spatial distribution on the surface of the measurement target by emitting laser pulses and resolving the reflected signals [10]. Furthermore, with its high accuracy and no need to touch the measurement target, 3D laser scanning has been widely used in the field of planimetric mass assessment [11], cadastral mapping [12], volume calculation [13], landslide monitoring [14], and 3D model reconstruction [15,16]. Previous research has relied on single surveying and mapping technology to reconstruct a 3D real-world model. This has resulted in 3D dam models that are inaccurate and incomplete. Consider the following examples: (1) Light and shadow effects lead to local model voids and texture dislocations; (2) data collection is difficult to implement in areas where UAV flying is prohibited and satellite positioning cannot be performed; and (3) blind spots for data collection are present because the dam surface environment is complex, roads are messy, and vehicle movement is frequent. In contrast, integrating the environmental information provided using 3D laser scanning and UAV tilt photography for 3D reconstruction can considerably improve the integrity, accuracy, and fidelity of the model.

Because UAV tilt photography and 3D laser scanning are two independent data acquisition systems, realising the deep integration of point clouds and images, integrating their advantages, and improving the consistency between 3D models and actual scenes are the research focus of multi-source 3D reconstruction. The traditional method is to export the 3D model reconstructed using UAV tilt photography to a point cloud format. Then, the same name marker points or marker surface cusps are arranged as control points under different data acquisition systems to complete the comprehensive sensing and deep integration of environmental information supported by point cloud compatibility. The current study has several limitations. The construction environment of the dam face is complex: Firstly, numerous mechanical vehicles, such as sprinklers, roller compactors, and bulldozers are present, and the stability of the marker points cannot be ensured. Secondly, manual participation is required owing to variations in dimension, scale, accuracy, and viewing angle among different data acquisition systems. This is time-consuming and labour-intensive and may introduce certain deviations [17]. Therefore, this paper aims to accomplish the unification of spatial relative positions of multimodal point cloud coordinate systems under different viewpoints and acquisition systems quickly and accurately from an algorithmic perspective. The most classic is the iterative closest point (ICP) algorithm [18], which is simple in principle and easy to operate but requires strict initial poses of the original point cloud and is prone to local optimality.

To resolve the abovementioned limitations, this paper proposes the use of a cross-source point cloud–trigonometric mutation chaotic Harris hawk optimisation (TMCHHO) fusion model for the 3D reconstruction of rockfill dams during construction. The key to this lies in (i) devising a means for collecting environmental information on the dam surface in a complex production environment as well as overcoming the phenomenon of data blindness and (ii) unifying the spatial coordinate system of multiple data sources in a fast and robust manner.

To achieve the first goal, an air–ground integrated environmental information sensing strategy based on multiple mapping systems is proposed. Furthermore, algorithms such as voxel filters and internal shape descriptors are used to realise the real-time acquisition and preprocessing of multi-source data. To achieve the second goal, an accurate fusion strategy based on multimodal data for rockfill dams during the construction period is proposed. In this method, the improved Harris hawk optimisation (TMCHHO) is integrated with the preliminary determination of the relative position of the point cloud in 3D space, and the ICP algorithm is incorporated to align the spatial positions accurately. Among them, the TMCHHO algorithm uses a piecewise linear chaotic map (PWLCM) sequence to generate a more randomly distributed initial population, and trigonometric mutation is applied to enhance the local chemotaxis ability [19,20]. Finally, considering the Lianghekou rockfill dam as an example, the reconstruction scheme of the 3D model of the dam surface is optimised. The distribution of Euclidean distance between aligned point clouds is used as an evaluation index to verify the fusion effect. Compared with the single surveying and mapping method, the effectiveness of the fusion model is verified.

## 2. Related Research

In the early 21st century, with the successful application of high-precision sensors, such as tilt photography cameras and LiDAR, to engineering construction, some scholars have conducted research on the fusion of cross-source data to achieve more realistic 3D model reconstruction. Escarcena et al. [21] devised a high-precision method for modelling ancient sites based on both photogrammetry and terrestrial laser scanning techniques. Kedzierski et al. [22] used different laser scanning systems to provide a database for high-precision urban model building. Nahon et al. [23] designed a collaborative measurement model of 3D laser scanning and UAV aerial surveys for terrain mapping dune corridors.

Some studies proposed cross-source point cloud registration algorithms that provide new ideas for the comprehensive sensing and accurate analysis of a construction site environment. According to the basic principle, cross-source point cloud registration can be broadly classified into two types: learning-based and optimisation-based methods. The learning-based registration method uses a neural network framework composed of multi-layer interconnected nodes to solve the point cloud spatial transformation matrix [24]. Aoki et al. [25] used PointNet as an imaging function to provide a new idea for point cloud registration. Lu et al. [26] generated key points based on learned matching probabilities of candidate sets. Yew et al. [27] replaced the space-based distance metric in the feature extraction stage of neural networks with hybrid feature-based distances to solve the problem of insufficient stability in the learning-based method. However, in learning-based registration algorithms, the design of transformation parameters is difficult to interpret, and often extensive experimentation is required to find the right architecture and hyperparameters. Moreover, the distribution of real-scene data cannot be too different from the training data [28,29]. An optimisation-based registration algorithm describes the process of point cloud registration as an optimisation problem. It mainly includes the variant algorithm of ICP, graph-based optimisation, and other methods. Yao et al. [30] queried the closest point based on the similarity matching of point cloud curve features. Serafin et al. [31] used normal vector and curvature to remove the points corresponding to errors after feature point matching and the angular difference in the normal vector of the corresponding points was also used as an additional error control term to enable fast computation.

However, previous research has not resolved the inadequacies of the traditional ICP algorithm, which has strict requirements on the initial position. Some scholars divide point cloud registration into two stages. The coarse registration stage roughly matches the spatial position of the point cloud, and the fine registration stage completes the precise alignment. Rusu et al. [32] proposed fast point feature histograms based on geometric relations to be used as descriptors of local features. Aiger et al. [33] proposed four-point congruent sets to solve low-overlap or noisy point cloud registration. Chua et al. [34] used point signatures to describe the local features of point clouds and applied them to automatic face recognition. In recent years, swarm intelligence optimisation algorithms simulating the laws of nature have been widely used in engineering. Some scholars have also attempted to achieve faster and more accurate point cloud registration using genetic [35] and particle swarm [36] algorithms. Experiments demonstrate that the HHO algorithm proposed by Heidari [37] performs better than other algorithms of the same classification. However, because it generates an initial population based on the random numbers of normal distribution, and the correlation among individuals in the population is ignored in a single iteration, the algorithm has a slow solution speed. Accordingly, this paper proposes a multi-strategy method for improved Harris hawk point cloud registration with strong robustness and high optimisation accuracy.

## 3. Research Framework

The research framework of the cross-source point cloud–TMCHHO fusion model for the high-precision, high-fidelity, and high-integrity 3D reconstruction of rockfill dams during construction is shown in Figure 1. It includes three components: data, method, and application layers.

Data layer: This part is based on multiple mapping systems to sense the environmental information of the dam during the construction period and produce multi-view tilt images from UAV tilt photography into a dense point cloud model. Other processes, such as noise filtering, point cloud simplification, and feature point extraction, are performed for dense and disordered point clouds to construct data layers as inputs.

Method layer: To achieve high-precision, high-realism, and high-integrity 3D reconstruction of rockfill dams, the cross-source point cloud–TMCHHO fusion model mainly consists of two parts. The first part determines the objective function and optimisation parameters according to the principle of point cloud space transformation in different coordinate systems. The second part combines the improved HHO algorithm with the PWLCM system and trigonometric mutation perturbation strategy to optimise algorithm performance.

Application layer: In this part, the strategy proposed in this paper is applied to a core-wall rockfill dam project, and the fusion accuracy is evaluated using the multimodal point cloud Euclidean distance distribution in the measurement area.

## 4. TMCHHO Fusion Model Based on Multimodal Point Cloud for Rockfill Dam

### 4.1. Cross-Source Point Cloud Registration Model

The cross-source point cloud registration model is described as a mathematical optimisation problem. Given two 3D point clouds with different coordinate systems, the optimal rigid transformation matrix is solved such that the relative spatial positions of the two clouds are aligned after the transformation. Let the mathematical expressions of the reference and target point clouds be as follows: $P_{s}=\{p_{is}\in R^{3},is=1,2,\ldots ,ms\}$ and $Q_{t}=\{q_{jt}\in R^{3},jt=1,2,\ldots ,nt\}$, where *ms* and *nt* are the number of data points that make up the point clouds, respectively. The conversion process can be described by the 3 × 3 rotation matrix (*R*) and 3D translation vector (*T*). They can be expressed by Equation (1), where *V_x_*, *V_y_*, and *V_z_* are the translation lengths along the direction of the spatial coordinate axis, and *θ_x_*, *θ_y_*, and *θ_z_* are the rotation angles around the spatial coordinate axis in the clockwise direction. The problem of multimodal point cloud spatial coordinate matching is abstracted as minimising the value of the F(R,T) function in Equation (1). The root mean square (RMS) value of corresponding points is used to characterise the accuracy of multimodal point cloud spatial matching.
(1)$$\left\{ \begin{array}{c} R=\left[\begin{array}{ccc} \cos\theta_{y}\cos\theta_{z} & \cos\theta_{y}\sin\theta_{z} & -\sin\theta_{y}\\ \sin\theta_{x}\sin\theta_{y}\cos\theta_{z}-\cos\theta_{x}\sin\theta_{z} & \sin\theta_{x}\sin\theta_{y}\sin\theta_{z}+\cos\theta_{x}\cos\theta_{z} & \sin\theta_{x}\cos\theta_{y}\\ \cos\theta_{x}\sin\theta_{y}\cos\theta_{z}+\sin\theta_{x}\sin\theta_{z} & \cos\theta_{x}\sin\theta_{y}\sin\theta_{z}-\sin\theta_{x}\cos\theta_{z} & \cos\theta_{x}\cos\theta_{y} \end{array}\right]\\ T=\left[\begin{array}{c} V_{x}\\ V_{y}\\ V_{z} \end{array}\right]\\ F(R,T)=\min{\left\|\left(RP_{s}+T)-Q_{t}\right)\right\|}^{2}\\ \mathrm{RMS}(P_{s},Q_{t})=\sqrt{\frac{1}{N_{S}}{\sum_{i,j\in N_{S}}\left\|(P_{s,i}+T)-Q_{t,j}\right\|}^{2}} \end{array}\right.$$
where *N_S_* is the number of data points contained in the intersection region of the multimodal point cloud.

### 4.2. Point Cloud Data Processing

Both 3D laser scanning and UAV tilt photography mapping generate massive quantities of data, which must be downsampled, and feature points should be extracted to eliminate the impact of overly numerous points on subsequent storage, transfer, and computation. The collected cross-source data may contain outliers affecting the subsequent feature point extraction and registration stages, and noise points must be filtered out. Hence, cross-source point cloud data preprocessing involves noise filtering, downsampling, and feature point extraction. For noise filtering, the typical radius filtering method is employed. In what follows, downsampling and feature point extraction are discussed, which are special processes for deriving cross-source point cloud characteristics.

Point cloud downsampling involves extracting the least number of data possible from areas with small curvatures and the greatest number of data possible from areas with large curvatures [38]. To simplify a point cloud swiftly without destroying its internal geometric features, a voxel filter is used. First, a 3D voxel grid coordinate system is created. Then, all the points inside each grid are replaced by the centroids inside each grid.

Feature points contain rich geometric feature information and can be used as an abstract representation of the original data shape. The point clouds of overlapping areas from different mapping systems and perspectives are relatively consistent. In this study, the intrinsic shape signature (ISS) [39] detection method is utilised to find feature points. The algorithm is easy to implement and yields stable results. Assume there exists a point cloud *P* containing *n_t_* data points, and the point coordinates are denoted by *p_m_*. The specific implementation steps are as follows:


(1)Define a local coordinate system for each point *p_m_* in the point cloud, *P*, and set the search radius, *d_iss_*, for the neighbourhood query.(2)Query all points in the region of *P* centred on *p_m_* with radius *d_iss_*. Calculate their weight values, *v_mn_*:(2)$$v_{mn}=\frac{1}{\left\|p_{m}-p_{n}\right\|},\left|p_{m}-p_{n}\right|<d_{iss}$$(3)Calculate the covariance matrix corresponding to each point *p_m_*:(3)$$cov(p_{m})=\frac{\sum_{\left|p_{m}-p_{n}\right|<d_{iss}}v_{mn}(p_{m}-p_{n}){\left(p_{m}-p_{n}\right)}^{T}}{\sum_{\left|p_{m}-p_{n}\right|<d_{iss}}v_{mn}}$$(4)Calculate the eigenvalues, $\left\{\lambda_{m}^{1},\lambda_{m}^{2},\lambda_{m}^{3}\right\}$, of each covariance matrix, and rank them from largest to smallest.(5)Set the threshold values $\varepsilon_{1}$ and $\varepsilon_{2}$. The points satisfying the two following inequalities are considered feature points and kept:(4)$$\left\{ \begin{array}{l} \frac{\lambda_{m}^{2}}{\lambda_{m}^{1}}<\varepsilon_{1}\\ \frac{\lambda_{m}^{3}}{\lambda_{m}^{2}}<\varepsilon_{2} \end{array}\right.$$

### 4.3. Traditional HHO Algorithm

The HHO algorithm is a gradient-free algorithm proposed in 2019 [37]. It solves the optimisation problem by simulating biological behaviours such as tracking, encircling, repelling, and attacking in the process of collaborative prey hunting by Harris hawk groups in nature, where each member of the Harris hawk population represents a candidate solution. The HHO algorithm consists of three phases: exploration, transition, and exploitation. When |*E_cv_*| ≥ 1, the Harris hawk algorithm performs a global exploration. In this stage, Harris hawks are randomly perched in a location, searching and tracking their prey through keen vision. The hawks select one of the two strategies with equal probabilities to determine the perching locations. When |*E_cv_*| < 1, the algorithm entered the exploitation phase, several Harris hawks swept up at the same time to form an encirclement and waited for the time of surprise attack. The HHO algorithm selects a suitable location update strategy based on two parameters: prey escape energy, *E_cv_*, and successful escape probability, *c*. Figure 2 is a simple example of the process through which a Harris hawk chases its prey. The whole search process of the algorithm is as follows:(5)$$\left\{ \begin{array}{c} \begin{array}{cc} E_{cv}=2E_{iv}(1-\frac{t_{s}}{T_{\max}}) & E_{iv}\in [-1,1] \end{array}\\ \begin{array}{cc} O(t_{s}+1)=\{\begin{array}{cc} \left(O_{prey}(t_{s})-O_{avg}(t_{s}))-e_{3}(L_{\min}+e_{4}(L_{\max}-L_{\min})\right) & q<0.5\\ O_{r}(t_{s})-e_{1}\left|O_{r}(t_{s})-2e_{2}O(t_{s})\right| & q\geq 0.5 \end{array} & \left|E_{cv}\right| \end{array}>1\\ \begin{array}{cc} \left\{ \begin{array}{c} \Delta O(t_{s})=O_{prey}(t_{s})-O(t_{s})\\ O(t_{s}+1)=\Delta O(t_{s})-E_{cv}\left|J_{r}O_{prey}(t_{s})-O(t_{s})\right| \end{array}\right. & c\geq 0.5,\left|E_{cv}\right|\geq 0.5 \end{array}\\ \begin{array}{cc} O(t_{s}+1)=O_{prey}(t_{s})-E_{cv}\left|\Delta O(t_{s})\right| & c\geq 0.5,\left|E_{cv}\right|\leq 0.5 \end{array}\\ \begin{array}{cc} \left\{ \begin{array}{c} P=O_{prey}(t_{s})-E_{cv}\left|J_{r}O_{prey}(t_{s})-O(t_{s})\right|\\ Q_{h}=P_{h}+S_{r}\times LF(D_{r})\\ O(t_{s}+1)=\left\{ \begin{array}{l} P_{h},f(P_{h})<f(O(t_{s}))\\ Q_{h},f(Q_{h})<f(O(t_{s})) \end{array}\right. \end{array}\right. & c<0.5,\left|E_{cv}\right|\geq 0.5 \end{array}\\ \begin{array}{cc} \left\{ \begin{array}{c} P_{h}=O_{prey}(t_{s})-E_{cv}\left|J_{r}O_{prey}(t_{s})-O_{avg}(t_{s})\right|\\ Q_{h}=P_{h}+S_{r}\times LF(D_{r})\\ O(t_{s}+1)=\left\{ \begin{array}{l} P_{h},f(P_{h})<f(O(t_{s}))\\ Q_{h},f(Q_{h})<f(O(t_{s})) \end{array}\right. \end{array}\right. & c<0.5,\left|E_{cv}\right|<0.5 \end{array} \end{array}\right.$$

In the above equation, *T*_max_ represents the maximum number of times that the population can updated; *E_iv_* denotes the initial escape energy corresponding to the algorithm when the prey perceives the crisis and starts to escape, which is a randomly distributed value on the range [−1, 1]; *E_cv_* is the escape energy value after the *t_s_*th iteration; *O_avg_*(*t_s_*) is the average value obtained after counting the positions of all population members; *O*(*t_s_*) is the position vector of the Harris hawk; *O_prey_*(*t_s_*) denotes the position of the prey at the *t_s_* th iteration; *L*_min_ and *L*_max_ are the lower and upper boundaries that constrain the space of values taken by the candidate solutions; *O_r_*(*t_s_*) represents the current location of a random Harris hawk individual; *q* is the control parameter of the global search policy; *q*, *e*_1_, *e*_2_, *e*_3_, and *e*_4_ are all random values distributed in the range of [0, 1]; Δ*O*(*t_s_*) represents the positional distance between the Harris hawk and its prey in the *t_s_*th iteration; *S_r_* is a two-dimensional vector composed of random numbers distributed on [0, 1]; *f*(·) denotes the adaptation value calculation for the current decision variable. *J_r_* is a random value distributed in the range of [0, 2]. *LF*(·) indicates that the parameters are updated using Lévy flight for long- and short-distance searches.

### 4.4. TMCHHO

An excellent swarm intelligence optimisation algorithm must maintain diversity when the population is initialised [40]. Thus, a global search is used to rapidly converge near the optimal solution after the optimisation task starts to execute, and the strong local convergence ability is maintained to fully explore the candidate solution space in its latter stage. The position vector at initialisation of the Harris hawk population is composed of normally distributed random numbers, thus increasing the time and computational costs of the global search. Furthermore, when the algorithm enters the development stage, the position update is prone to stagnation owing to the lack of learning among individuals of the population in a single iteration. This affects the accuracy of the algorithm for finding the best solution. To resolve these problems, the two following improvements are introduced:
(1)Mapping the PWLCM sequence to optimise the distribution of the initial population:

A reasonable initial solution for the population can increase the search space and accelerate the convergence of the population. Chaos [41], as a nonlinear phenomenon in nature, has been widely used in swarm intelligence optimisation algorithms because of its ergodicity, randomness, and sensitivity to initial conditions. Presently, logistic chaotic maps are widely used in metaheuristic algorithms. Previous studies emphasise that PWLCM systems have unique advantages compared with logistic maps [42,43]. Figure 3a–d show the histograms and point distribution diagrams of the logistic map and PWLCM system, respectively.

Figure 3 indicates that the probability of the logistic sequence in [0, 0.1] and [0.9, 1] is higher than that in other intervals. The PWLCM sequence exhibits better randomness and ergodicity. In this study, the PWLCM system is mapped into the value space of the optimisation variables, thus replacing the pseudo-random numbers in initialising the population of Harris hawks and reducing the influence of insufficient population diversity on the global exploration capacity. The expression for the PWLCM system mentioned above is as follows:(6)$$r_{k+1}=\{\begin{array}{cc} \frac{r_{k}}{t_{cr}} & r_{k}\in [0,t_{cr})\\ \frac{r_{k}-t_{cr}}{0.5-t_{cr}} & r_{k}\in (t_{cr},0.5]\\ \frac{1-t_{cr}-r_{k}}{0.5-t_{cr}} & r_{k}\in (0.5,1-t_{cr}]\\ \frac{1-r_{k}}{t_{cr}} & r_{k}\in (1-t_{cr},1] \end{array}$$

In the above formula, the value range of the control parameter *t_cr_* is (0, 0.5), which is set to 0.4 in this paper; k=1,2,…,n; and *r_k_* and *r_k_*_+1_ are the chaotic numbers in the chaotic sequence.

(2)Introduction of trigonometric mutation to improve location update strategies:

Trigonometric mutation [44] refers to randomly extracting three mutually different solutions, i.e., *Z_t_*_1,*n*_, *Z_t_*_2,*n*_, and *Z_t_*_3,*n*_, as vertices in the population. Moreover, the vector differences, i.e., $\left(Z_{t1,n}-Z_{t2,n}\right)$, $\left(Z_{t2,n}-Z_{t3,n}\right)$, and $\left(Z_{t3,n}-Z_{t1,n}\right)$, are used as the side lengths of the hypergeometric triangular search space. Three fitness values *f*(*Z_t_*_1,*n*_), *f*(*Z_t_*_2,*n*_), and *f*(*Z_t_*_3,*n*_) are used to perturb the stagnant population. In the development stage of the Harris hawk algorithm, a trigonometric mutation is introduced to control the population to always mutate toward the individual with the optimal objective function value and adaptively adjust the forward progress of each Harris hawk to improve the learning ability among the individuals of the population in each iteration. The equation for trigonometric mutations is derived as follows:(7)$$\left\{ \begin{array}{c} K=f(Z_{t1,n})+f(Z_{t2,n})+f(Z_{t3,n})\\ \left\{ \begin{array}{l} k_{1}=f(Z_{t1,n})/K\\ k_{2}=f(Z_{t2,n})/K\\ k_{3}=f(Z_{t3,n})/K \end{array}\right.\\ \left\{ \begin{array}{l} td_{1}=(k_{2}-k_{1})\times (Z_{t1,n}-Z_{t2,n})\\ td_{2}=(k_{3}-k_{2})\times (Z_{t2,n}-Z_{t3,n})\\ td_{3}=(k_{1}-k_{3})\times (Z_{t3,n}-Z_{t1,n}) \end{array}\right.\\ v_{mn}=(Z_{t1,n}+Z_{t2,n}+Z_{t3,n})/3+or_{m,n}\times (td_{1}+td_{2}+td_{3}) \end{array}\right.$$

In the above, *f*(*Z_tm,n_*) represents the objective function value of the current individual; *K* is the sum of the objective function values of the three extracted solutions; *k*_1_, *k*_2_, and *k*_3_ are the ratios of the corresponding function values; *td*_1_, *td*_2_, and *td*_3_ represent the forward progress along the directions of three individuals, respectively; and *or_m_*_,*n*_ is the Gaussian distribution scaling factor, that is, *or_m_*_,*n*_~*N*(0, *β*). In this paper, *β* is set to 0.1.

The centre point of the hypergeometric triangle search space is used as the base vector position, and the individual mutation direction and step size are controlled by the weighted vector. When $k_{m}-k_{n}<0$, the base vector moves towards individual *Z_m_*. When $k_{m}-k_{n}>0$, the base vector moves towards individual *Z_n_*, ensuring that the candidate solution updates its position in the direction of the better individual at each iteration. As shown in Figure 4, when $k_{3}<k_{2}<k_{1}$, the individual moves in direction *Z_t_*_3,*n*_, which has the smallest objective function value. Furthermore, the Gaussian distribution factor, *or_m_*_,*n*_, regulates the degree of trigonometric mutation perturbation. Figure 5 shows the flowchart of the TMCHHO algorithm.

### 4.5. ICP Algorithm

After rough registration, the point cloud coordinate system is roughly aligned, but it still cannot meet the needs of engineering. It is necessary to use the ICP algorithm to further improve the registration accuracy. The specific implementation steps are as follows:
(1)Find the corresponding point (*Q_i_*) of each point (*P_i_*) in the source point cloud (*P_S_*) in the target point cloud (*Q_t_*).(2)Calculate *D_dis_* and solve the spatial transformation parameters using the singular value decomposition method.
(8)$$D_{dis}=\frac{1}{N}\sum_{i=1}^{N}{\left\|Q_{i}-(RP_{i}+T)\right\|}^{2}$$(3)The spatial transformation parameters obtained in the previous step are applied to *P_S_*, and the new point cloud obtained is named *P_n_*.(4)Use *P_n_* and *Q_t_* to calculate the target value (*D_dis_*) in Equation (8). If *D_dis_* is less than a certain threshold, stop the iteration; otherwise, the algorithm proceeds to the next iteration and repeats steps (1)–(3).

## 5. Simulation Experiments and Engineering Application

This section contains three parts; firstly, the robustness and accuracy of the TMCHHO algorithm are verified based on optimisation experiments using benchmark functions. In addition, the feasibility of the TMCHHO algorithm to solve the point cloud coordinate system unification problem is verified based on registration experiments using a standard point cloud dataset. Finally, the TMCHHO point cloud alignment algorithm is applied to the massive multimodal point clouds generated during the construction of the Lianghekou project, demonstrating the improved accuracy and completeness of the fusion model compared with the existing single reverse modelling approach.

### 5.1. Benchmark Function Optimisation Experiments

In order to verify the superiority of the TMCHHO algorithm, seven representative benchmark functions were selected to carry out optimisation experiments, as shown in Table 1, where *f*_min_ denotes the optimal value at which the benchmark function can converge theoretically, and range is the constraint on the candidate solution. The functions [*f*_1_(*x*) − *f*_4_(*x*)] are single-peaked benchmark functions with only one local extreme value, which are suitable for testing the global exploration ability. The functions [*f*_5_(*x*) − *f*_7_(*x*)] are complex multi-peaked benchmark functions with multiple local extreme values, which are suitable for testing the ability to jump out of premature convergence.

To ensure the scientific nature of the experiments, 10 control groups were set up. Therefore, we chose the widely concerned whale optimisation algorithm (WOA), particle swarm optimisation (PSO), butterfly optimisation algorithm (BOA), the traditional HHO algorithm, and the dingo optimisation algorithm proposed in 2021 [45] to conduct experiments when the decision variable dimensions were 10 and 50, respectively. Additionally, the performance of the algorithms was analysed based on the pairs of experimental results for each test group. The control parameters of each algorithm are shown in Table 2, and the statistical results of the optimisation experiments of each group are shown in Table 3.

Regardless of whether the dimension was 10 or 50, the TMCHHO algorithm could approach the global solution on the seven benchmark test functions. Among them, the results of convergence on the *f*_1_(*x*), *f*_2_(*x*), *f*_3_(*x*), *f*_4_(*x*), and *f*_6_(*x*) functions were close to the theoretical optimal value, and in the complex multi-peaked functions *f*_5_(*x*) and *f*_7_(*x*), the experimental results of the TMCHHO algorithm could reach the theoretical optimal value. Although the traditional HHO algorithm performed comparably to the TMCHHO algorithm on *f*_3_(*x*), *f*_5_(*x*), and *f*_7_(*x*), the performance of the TMCHHO algorithm on the other four functions was the best among the six algorithms, indicating that the use of trigonometric mutation and the PWLCM system led to a significant improvement in the ability of the HHO algorithm to seek the optimal solution and robustness.

Figure 6 illustrates the variation curves of the fitness values of the six algorithms on the single-peaked test functions *f*_1_(*x*) and *f*_2_(*x*), and the multi-peaked test functions *f*_5_(*x*) and *f*_6_(*x*), when the dimensionality of the decision variables was 10. As can be seen from the graphs, the improved Harris hawk algorithm converged faster and had a higher optimisation-seeking accuracy than other population intelligence optimisation algorithms.

### 5.2. Standard Point Cloud Dataset Registration Experiments

To verify the feasibility of the TMCHHO algorithm in the field of point cloud registration, experiments were conducted using the data obtained from the Armadillo, Bunny, Dragon, and Happy datasets, and they were compared with the data from other algorithms. Figure 7 shows the standard point cloud set used in this paper. Following the single-variable principle of comparative experiments, the same population size, iteration times, and initial value range were set for each swarm intelligence optimisation algorithm. In the experiments presented in this section, the number of iterations, population size, and initial value range were set as 500, 30, and [−50, 50], respectively. All algorithms were programmed in MATLAB R2016b, and an Intel Core i7-9750H 32 G computer was used.

Point cloud data *P* and *Q* from different perspectives were set as the operation objects, and the spatial position alignment of the two point clouds was set as the optimisation goal. Accordingly, various swarm intelligence optimisation algorithms were employed to conduct 20 experiments on multi-point cloud sets.

The point clouds were first processed using different methods, such as voxel filtering and ISS feature point extraction. The reasonable values of *r*, *ε*_1_, and *ε*_2_ (voxel filtering and feature point extraction parameters) could reduce computational effort as well as preserve its inherent geometric feature information. Based on experience, and after several experiments, the values of *r*, *ε*_1_, and *ε*_2_ were set to 0.001, 0.65, and 0.6, respectively.

In this section, we use the RMS values mentioned in Section 4.1 to characterise the alignment accuracy of the two point clouds. In order to observe the results of the point cloud registration experiments more intuitively, Figure 8 plots the variation curves of RMS values during the iterations of the six algorithms.

As shown in Figure 8, the iterative curves of each optimisation algorithm had a downward trend as the number of population update times increased. In contrast, the TMCHHO algorithm had a relatively large RMS value in the first iteration because of the more dispersed initial population; however, the reduction rate was rapid in the early stage. After more than 100 iterations, all the algorithms exhibited a stable trend; however, the convergence value of the TMCHHO algorithm was smaller after 500 iterations, proving the superiority of the algorithm.

To analyse the performance of the TMCHHO algorithm further, Table 4 outlines the experimental results of each algorithm on different datasets. It can be seen from the table that the performance of the TMCHHO algorithm on the two data indicators of the average and worst values was the best among the six algorithms, and its stability performance was second only to the traditional HHO algorithm and better than the WOA, BOA, DOA, and PSO algorithms. Compared with the traditional HHO algorithm, the registration accuracy on the Armadillo, Bunny, Dragon, and Happy datasets improved by 52.11%, 72.91%, 48.51%, and 77.69%, respectively. Figure 9 shows a boxplot that intuitively reflects the distribution and average level of each algorithm. The TMCHHO algorithm achieved satisfactory results for the coordinate matching problem on the above four standard point cloud datasets. Other comparative analysis algorithms were prone to falling into local loops or having inadequate stability, resulting in unsatisfactory final convergence results. The above statistical data demonstrate the feasibility and superiority of the TMCHHO application in the field of point cloud registration.

As shown in Figure 10, only the ICP algorithm was used for registration, which was greatly affected by the initial pose of the point cloud, and the registration effect was not ideal. The use of the TMCHHO coarse registration algorithm overcame this defect. Table 5 also shows that the combination of TMCHHO and ICP algorithms can significantly improve registration accuracy.

### 5.3. Engineering Application

The Lianghekou rockfill dam project was used as the research object. The crest of the dam is 668.7 m long and 16 m wide. Firstly, information on the complex environment of the dam surface during the construction period was collected using multiple mapping systems. Secondly, the TMCHHO algorithm was used to complete the unification of multimodal point cloud spatial locations. Finally, the Euclidean distance distribution of the measurement area was used to evaluate the point cloud fusion accuracy.

The construction environment of the dam surface is complex, and vehicles such as trucks, vibratory rollers, and sprinklers operate continuously, rendering continuous measurement difficult to implement. Therefore, separate 3D laser scanning processes cannot collect the complete environmental information of the dam surface and surrounding terrain. By contrast, UAV tilt photography is less affected by the ground construction environment and has flexible air-to-ground observation capability, but its limitation is that some spatial information at the bottom and side of the feature may be missing or deformed. Figure 11 shows a schematic of the data acquisition strategy, which adopts an integrated ground-to-air strategy to collect point clouds and images to overcome the blind area phenomenon and reduce labour costs.

The equipment used for point cloud data acquisition was a Faro Focus^S^ 350 terrestrial 3D laser scanner. During the field acquisition, we surveyed the measurement area in advance, adjusted the station location according to the actual scanning situation, ensured an overlap rate of at least 30% among adjacent stations, and set the scanning range to 360° × 300°. A total of 22 stations were deployed for this scan. Figure 12a shows the acquisition device, and Figure 12b shows the acquired point cloud data.

For image collection, DJI Genie UAV was employed. To ensure the safe flight and high-quality data acquisition of the UAV, the route was planned in advance, the heading and side overlap rates were set to 80%, and the UAV was manually controlled to take supplementary shots in local areas. In the postprocessing stage, the supporting software Context Capture was used to produce the collected multi-view oblique images into a 3D dense point cloud model and output it in *.las* format, as shown in Figure 13b. The slope part was cropped and used as the data source for solving the transformation matrix.

After entering the data processing stage, downsampling and feature point extraction were performed for massive multimodal point clouds with disorder characteristics, where the edge length of the voxel grid was set to 0.1, and the spherical search radius in the internal shape descriptor algorithm was set to 0.3. Table 6 lists the number of data points in the point cloud before and after downsampling. In this study, the fusion effect of the TMCHHO algorithm was verified based on the massive geometric spatial information provided by multiple mapping systems during the construction session of the rockfill dam.

Using six algorithms to register multimodal point clouds in practical engineering, the obtained registration accuracy histogram is shown in Figure 14a. It can be seen from the figure that the TMCHHO algorithm performed best, with an RMS value of 1.01. Figure 14b shows the RMS value variation curve when applying the six algorithms to multimodal point clouds. The WOA, PSO, BOA, and DOA algorithms converged at 200 iterations, while the HHO and TMCHHO algorithms were superior in the second half of the iterative process. The fitness value obtained using the TMCHHO algorithm was the smallest, and the rigid transformation matrix obtained with this algorithm was used as the final application scheme. Figure 15 shows the process of spatial coordinate system matching for a multimodal point cloud with an unknown orientation pose. It can be seen from the figure that the use of the TMCHHO registration algorithm enabled the source point cloud to also obtain a good initial pose. The two point clouds were completely aligned after applying the ICP fine registration algorithm.

The conventional evaluation method is used to lay control points of the same name under multiple measurement techniques, use a total station to collect geodetic coordinates, and compare the absolute accuracy of model 3D reconstruction [23]. However, the coordinate acquisition of control points cannot be guaranteed due to the blind area phenomenon caused by the complex construction environment of the dam face and the special topographic conditions of the rockfill dam, so we performed a relative comparison of the dense point cloud models produced using the two measurement techniques. As shown in Figure 16, the dense point clouds produced using UAV tilt photography and 3D laser scanning were defined as the reference model and the target model, respectively, and a colour map with the Euclidean distance of the measured area was considered the index. The closer the colour was to red, the greater the deviation of the corresponding point cloud. To further evaluate the multimodal point cloud fusion, Table 7 lists the various indicators of Euclidean distances of the multimodal point cloud in the five measurement areas. Figure 17 shows the histogram of the Euclidean distance of the point cloud within the five measurement areas. Among them, the point cloud deviations were most distributed in the range of [0, 0.06], the average deviation range was 1.8–3.1 cm, and the degree of agreement between the two models was high. The intercepted point cloud data can be used to complement the missing parts. For example, the missing point cloud data of UAV tilt photography in the box shown in Figure 18a,c can be filled by a point cloud derived from 3D laser scanning. Figure 18b,d show the fusion point cloud model in different views.

## 6. Conclusions

The accuracy, completeness, and authenticity of the 3D model reconstruction of the dam during the construction period determine its consistency with the actual project. In the traditional technology involving the 3D reconstruction of dams, the simultaneous acquisition of high-precision 3D information and texture features of real ground objects from multiple angles is difficult to achieve. Accordingly, a TMCHHO fusion model for the 3D reality modelling of rockfill dams during construction based on multi-source point clouds was developed and validated based on actual projects. The following findings were observed:

(1) An air–ground comprehensive perception system based on multiple surveying and mapping systems was proposed, which realises the real-time acquisition of point clouds and images containing considerable amounts of spatial and texture information of rockfill dams during construction. The collected multi-source data were preprocessed using a voxel filter and ISS descriptors. (2) A high-precision, high-integrity, and high-fidelity TMCHHO point cloud fusion model was developed for the 3D reconstruction of rockfill dams during dam construction. The model integrates the improved Harris hawk coarse registration and ICP fine registration algorithms. The former combines the PWLCM system in the initialisation stage to reduce the impact of initial values on global exploration. In the local development stage, a trigonometric mutation was introduced to perturb the population individuals such that falling into local optima can be avoided. (3) The TMCHHO algorithm was verified to achieve better accuracy than the BOA, PSO, WOA, and HHO using standard point cloud datasets (i.e., Armadillo, Bunny, Dragon, and Happy) in point cloud registration experiments. (4) The method proposed in this paper was applied to the Lianghekou rockfill dam, and the improvement in model integrity compared with a single surveying and mapping system was verified.

The results of this study can automatically achieve the fusion of multimodal point clouds, saving labour time costs while improving the integrity of the model. Accordingly, the study results contribute to advancing the intelligent development of rockfill dam construction management and decision making. This study can be extended and applied to other hydropower construction projects during their construction period in the realistic 3D modelling of cross-source data, which have certain universality. In addition, owing to the multiple data format changes involved in the fusion modelling process, in the future, the authors intend to conduct research on cross-source point-cloud-integrated data acquisition and processing, as well as fused big data processing platforms, to achieve highly accurate and stable engineering application targets.

## Figures and Tables

**Figure 1 sensors-23-04942-f001:**
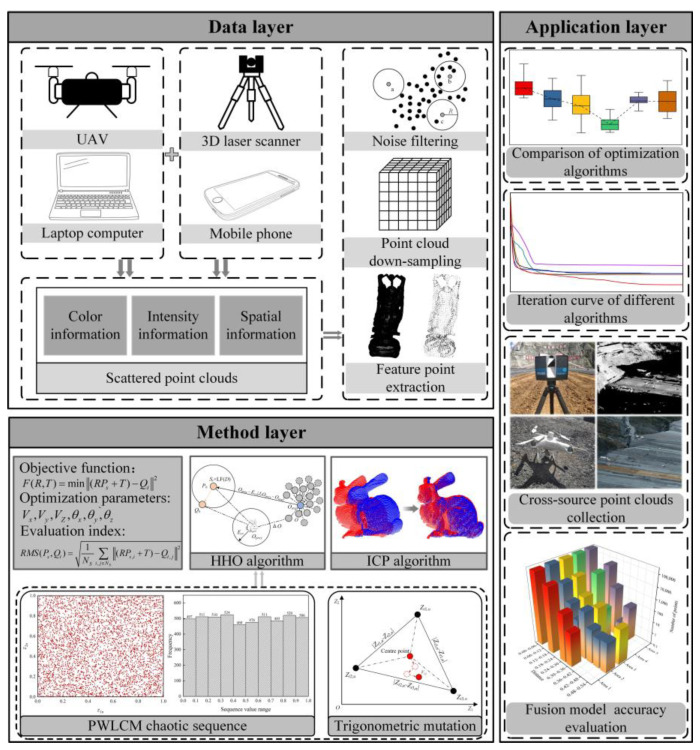
Research framework diagram.

**Figure 2 sensors-23-04942-f002:**
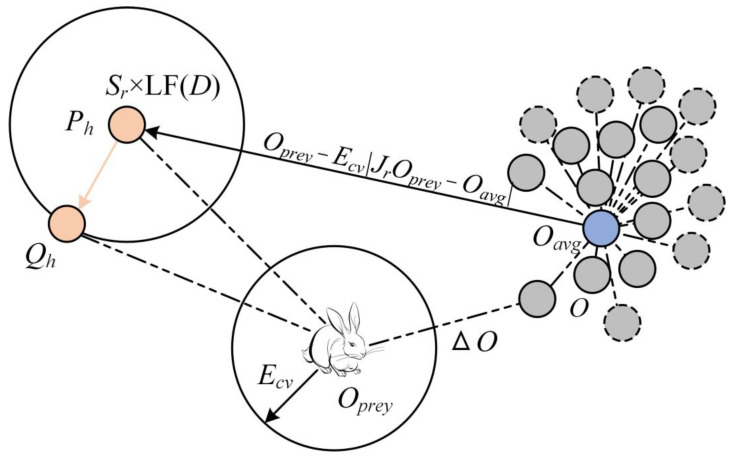
The process of Harris hawk chasing its prey.

**Figure 3 sensors-23-04942-f003:**
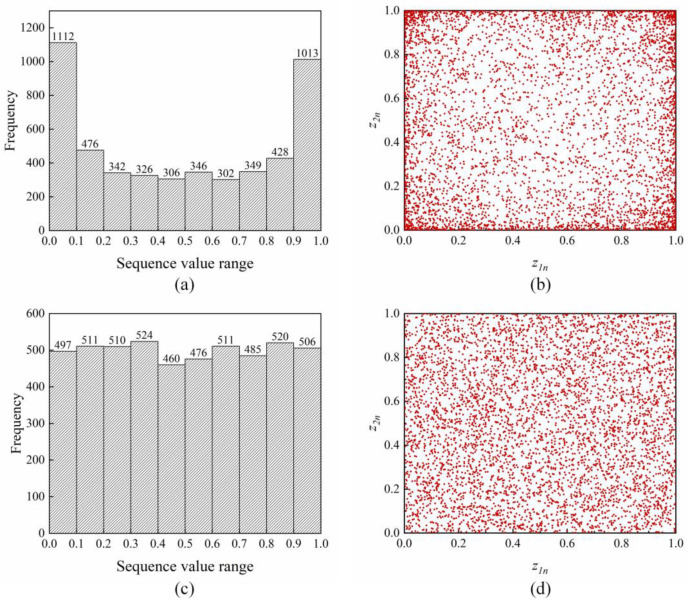
(**a**) Histogram of the logistic sequence; (**b**) distribution point diagram of the logistic sequence; (**c**) histogram of the PWLCM sequence; (**d**) distribution point diagram of the PWLCM sequence.

**Figure 4 sensors-23-04942-f004:**
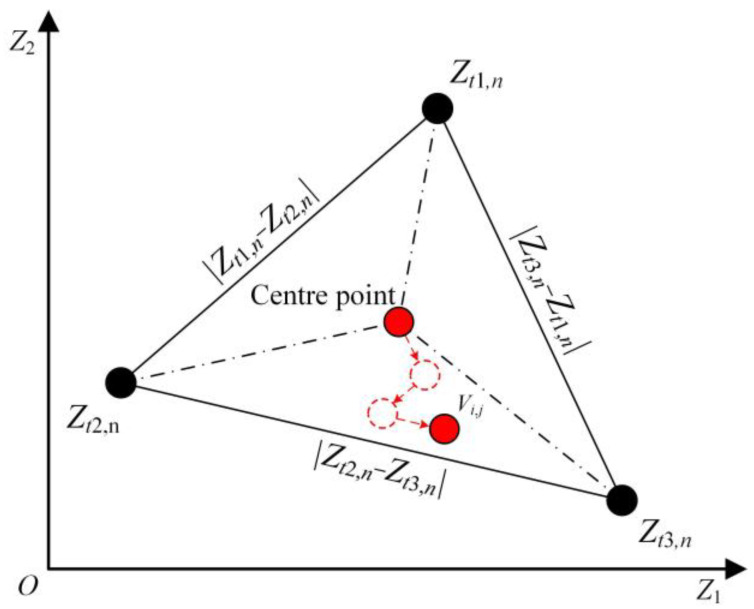
The progress of trigonometric mutation.

**Figure 5 sensors-23-04942-f005:**
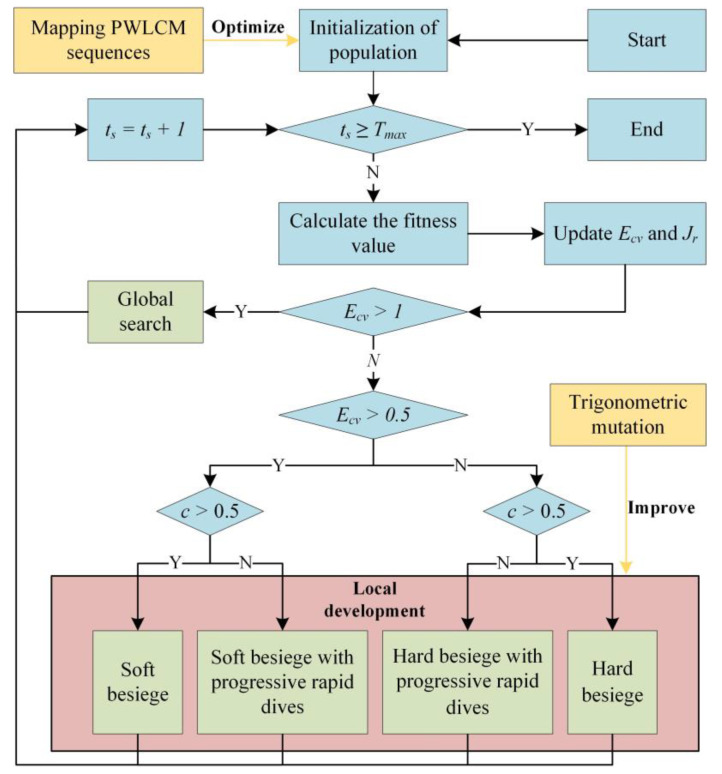
Flowchart of TMCHHO algorithm.

**Figure 6 sensors-23-04942-f006:**
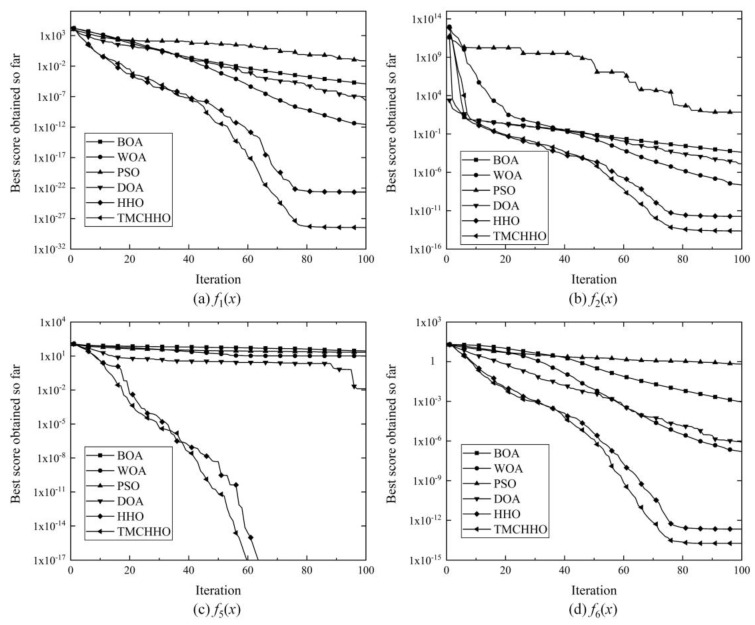
Comparison of algorithm optimisation ability: (**a**) *f*_1_(*x*); (**b**) *f*_2_(*x*); (**c**) *f*_5_(*x*); (**d**) *f*_6_(*x*).

**Figure 7 sensors-23-04942-f007:**
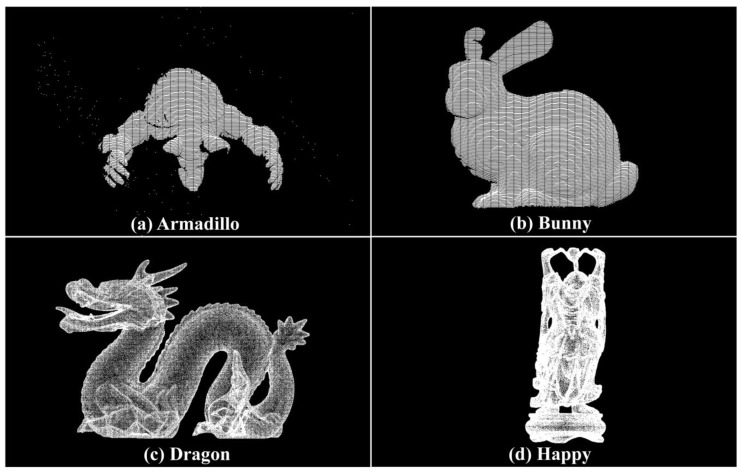
Standard point cloud datasets: (**a**) Armadillo; (**b**) Bunny; (**c**) Dragon; (**d**) Happy.

**Figure 8 sensors-23-04942-f008:**
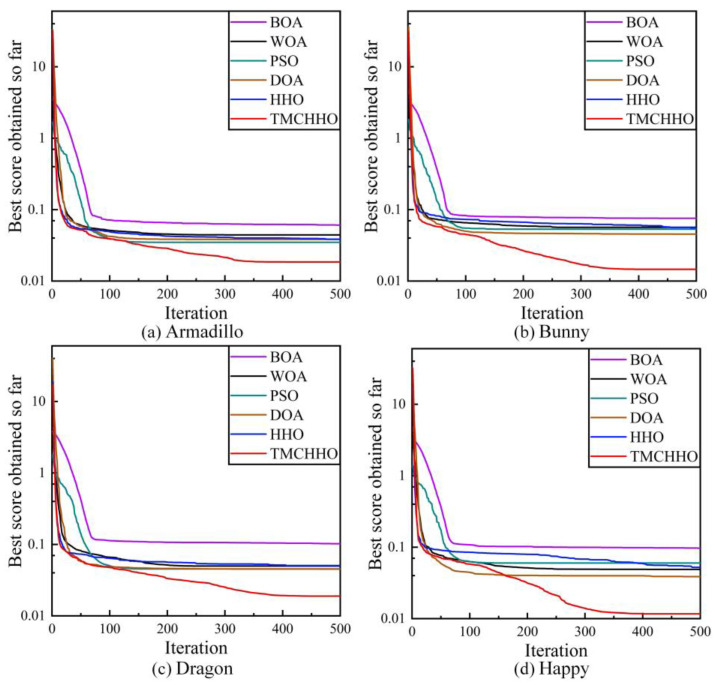
Iteration curves of each algorithm on different datasets: (**a**) Armadillo dataset; (**b**) Bunny dataset; (**c**) Dragon dataset; (**d**) Happy dataset.

**Figure 9 sensors-23-04942-f009:**
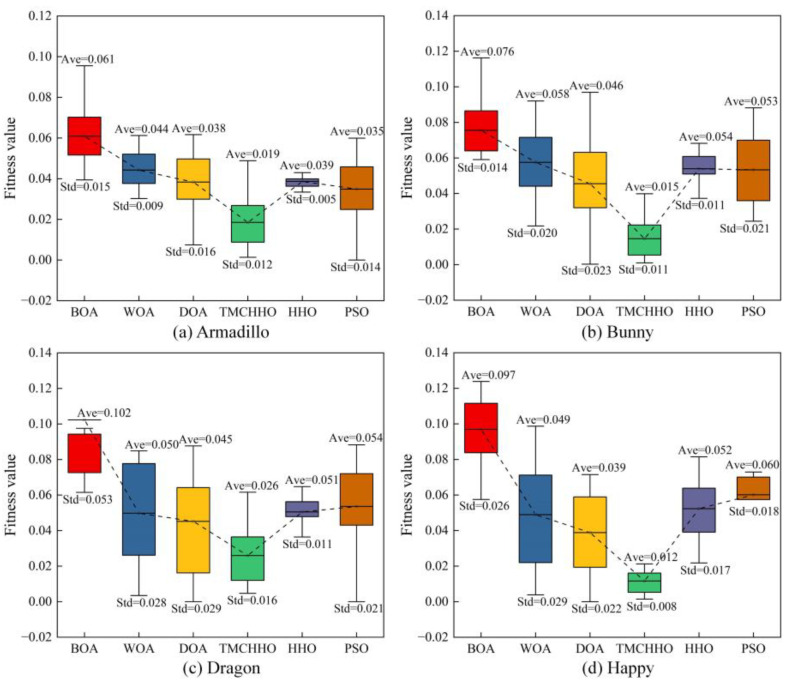
Comparison of algorithm alignment effects: (**a**) Armadillo dataset; (**b**) Bunny dataset; (**c**) Dragon dataset; (**d**) Happy dataset.

**Figure 10 sensors-23-04942-f010:**
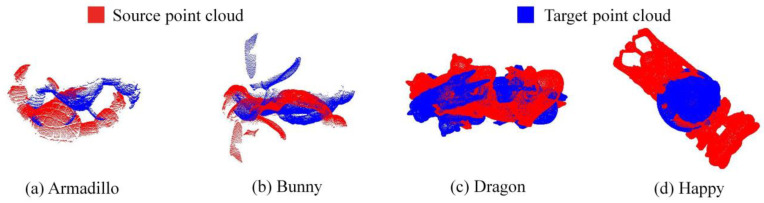
Alignment results using the ICP algorithm only.

**Figure 11 sensors-23-04942-f011:**
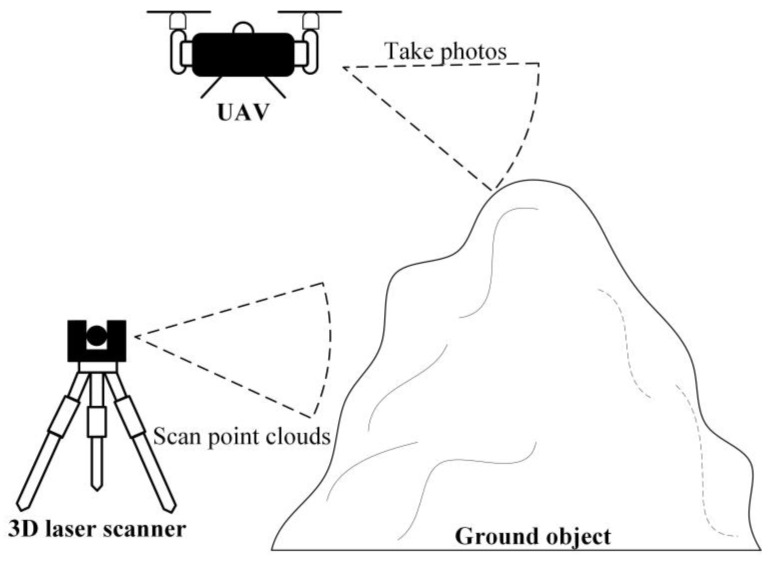
Schematic diagram of data acquisition strategy.

**Figure 12 sensors-23-04942-f012:**
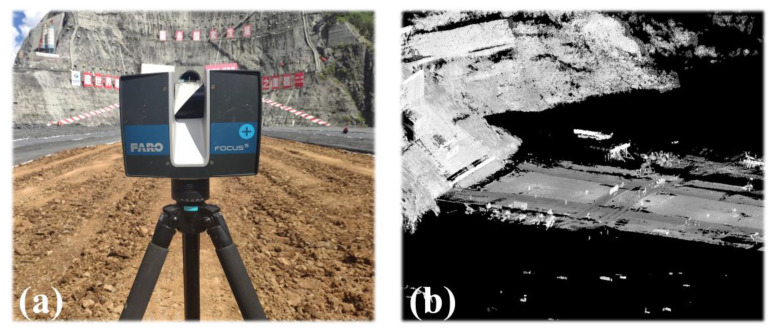
(**a**) Terrestrial laser scanner; (**b**) Laser scanning point cloud.

**Figure 13 sensors-23-04942-f013:**
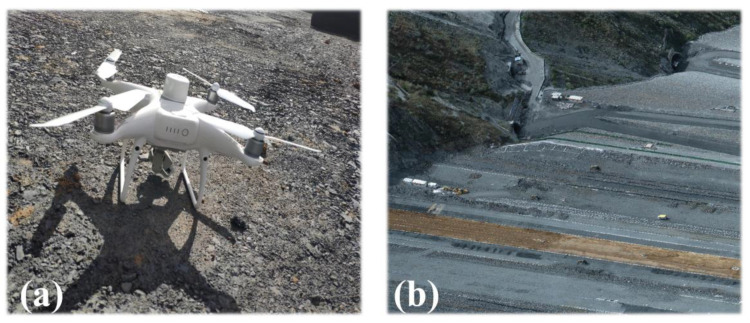
(**a**) UAV tilt photography; (**b**) point cloud from UAV tilt photography.

**Figure 14 sensors-23-04942-f014:**
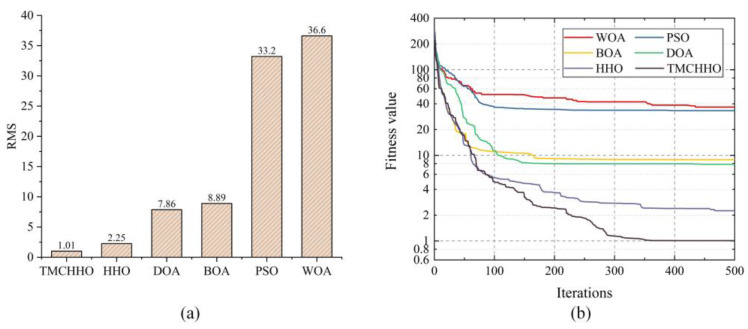
(**a**) Registration accuracy of different algorithms; (**b**) variation in fitness values in the actual project.

**Figure 15 sensors-23-04942-f015:**
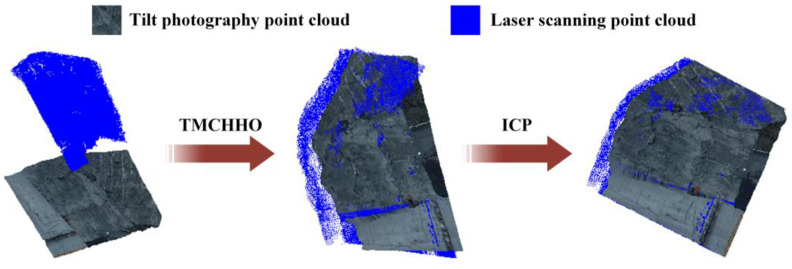
The process of cross-source point cloud fusion.

**Figure 16 sensors-23-04942-f016:**
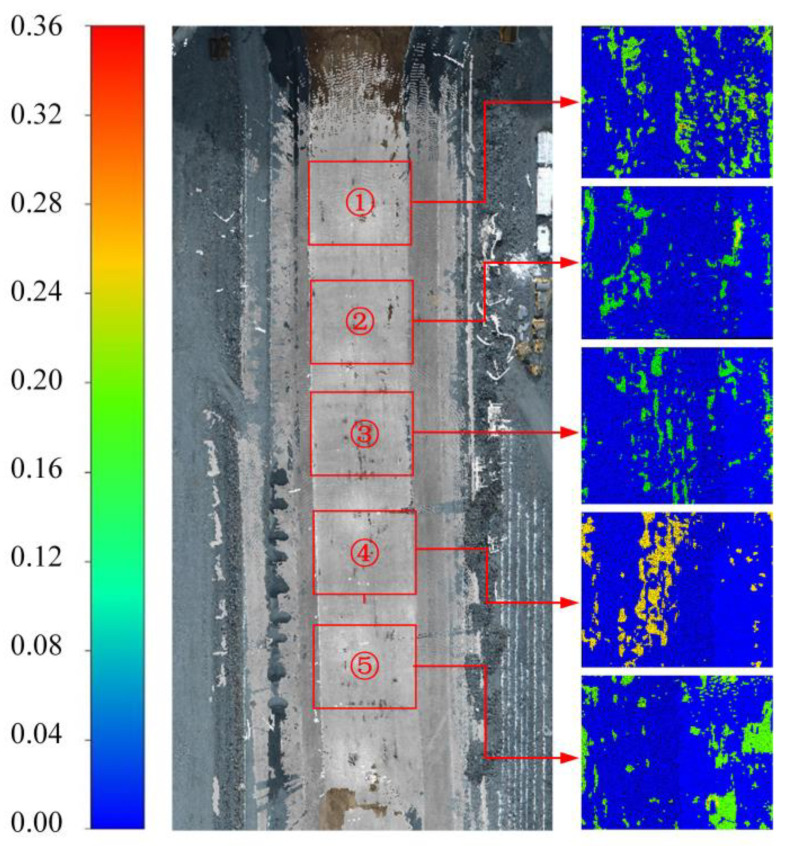
Colour mapping diagram of the survey area.

**Figure 17 sensors-23-04942-f017:**
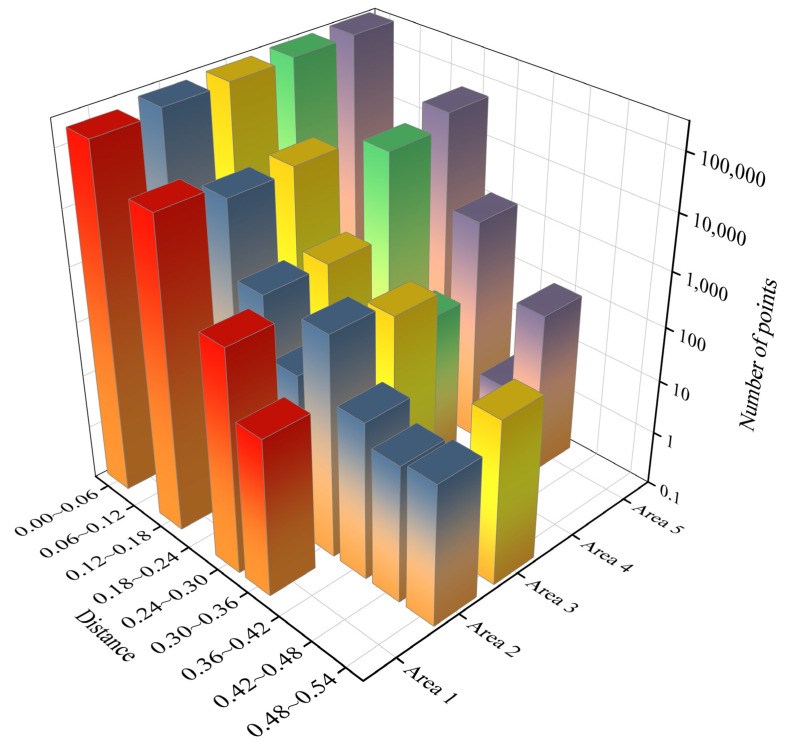
Evaluation of fusion model accuracy.

**Figure 18 sensors-23-04942-f018:**
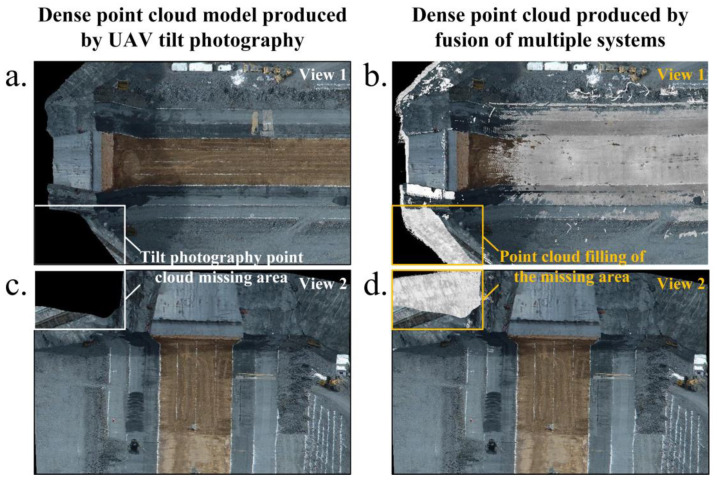
(**a**) Tilt photography point cloud under the top-down view along the dam axis; (**b**) fusion point cloud under the top-down view along the dam axis; (**c**) tilt photography point cloud under the 30-degree overhead view perpendicular to the dam axis; (**d**) fusion point cloud under the 30-degree overhead view perpendicular to the dam axis.

**Table 1 sensors-23-04942-t001:** Benchmark functions.

Expressions	Dimensions	*f* _min_	Range
$f_{1}(x)=\sum_{i=1}^{n}x_{i}^{2}$	10/50	0	[−100, 100]
$f_{2}(x)=\sum_{i=1}^{n}\left|x_{i}\right|+\prod_{i=1}^{n}\left|x_{i}\right|$	10/50	0	[−10, 10]
$f_{3}(x)=\sum_{i=1}^{n}{\left(\sum_{j-1}^{i}x_{j}\right)}^{2}$	10/50	0	[−100, 100]
$f_{4}(x)=\max_{i}\left\{\left|x_{i}\right|,1\leq i\leq n\right\}$	10/50	0	[−100, 100]
$f_{5}(x)=\sum_{i=1}^{n}\left[x_{i}^{2}-10\cos(2\pi x_{i})+10\right]$	10/50	0	[−5.12, 5.12]
$f_{6}(x)=-20e^{-0.2\sqrt{\frac{1}{n}\sum_{i=1}^{n}x_{i}^{2}}}-e^{\frac{1}{n}\sum_{i=1}^{n}\cos(2\pi x_{i})}+20+e$	10/50	0	[−32, 32]
$f_{7}(x)=\frac{1}{4000}\sum_{i=1}^{n}x_{i}^{2}-\prod_{i=1}^{n}\cos(\frac{x_{i}}{\sqrt{i}})+1$	10/50	0	[−600, 600]

**Table 2 sensors-23-04942-t002:** Parameters settings.

Algorithms	Parameters	Values
TMCHHO	Control parameter in Levy flight	1.5
BOA	Stimulus intensity	0.01
	Power exponent	0.1
PSO	Social constant	1
	Local constant	0.8
WOA	Spiral shape constant	1
HHO	Control parameter in Levy flight	1.5
DOA	Probability of hunting or scavenger	0.5
	Probability of group or persecution attack	0.7

**Table 3 sensors-23-04942-t003:** The statistical data of testing results (Note: The best results are highlighted in bold).

Functions	Algorithms	Experimental Results (Dimensions Are 10/50)			
		Ave	Best	Worst	Std
*f*_1_(*x*)	BOA	1.09 × 10^−5^/1.88 × 10^−5^	6.70 × 10^−06^/1.42 × 10^−5^	1.44 × 10^−5^/2.48 × 10^−5^	1.86 × 10^−6^/2.51 × 10^−6^
	WOA	1.65 × 10^−12^/8.64 × 10^−11^	6.19 × 10^−16^/1.14 × 10^−14^	2.62 × 10^−11^/1.41 × 10^−9^	5.35 × 10^−12^/2.67 × 10^−10^
	PSO	3.09 × 10^−2^/1.41 × 10^4^	1.48 × 10^−12^/1.86 × 10^03^	9.27 × 10^−1^/4.73 × 10^4^	1.69 × 10^−1^/9.70 × 10^3^
	DOA	2.10 × 10^−8^/6.75 × 10^−9^	**3.27 × 10^−84^/1.90 × 10^−97^**	6.22 × 10^−7^/1.33 × 10^−7^	1.13 × 10^−7^/2.55 × 10^−8^
	HHO	2.34 × 10^−22^/4.18 × 10^−22^	6.90 × 10^−32^/6.90 × 10^−33^	5.74 × 10^−21^/1.14 × 10^−20^	1.06 × 10^−21^/2.09 × 10^−21^
	TMCHHO	**3.46 × 10^−29^/8.89 × 10^−26^**	8.36 × 10^−37^/6.92 × 10^−37^	**5.27 × 10^−28^/1.87 × 10^−24^**	**1.04 × 10^−28^/3.59 × 10^−25^**
*f*_2_(*x*)	BOA	4.01 × 10^−4^/7.81 × 10^72^	5.49 × 10^−6^/1.36 × 10^69^	9.43 × 10^−4^/8.70 × 10^73^	2.84 × 10^−4^/2.02 × 10^73^
	WOA	1.83 × 10^−8^/3.63 × 10^−7^	1.54 × 10^−10^/4.84 × 10^−10^	1.34 × 10^−7^/7.95 × 10^−6^	3.30 × 10^−8^/1.45 × 10^−6^
	PSO	6.98 × 10^1^/1.75 × 10^46^	1.13 × 10^−5^/1.75 × 10^3^	2.85 × 10^2^/5.25 × 10^47^	8.64 × 10/9.58 × 10^46^
	DOA	1.22 × 10^−5^/4.45 × 10^−5^	**1.82 × 10^−42^/3.13 × 10^−44^**	2.41 × 10^−4^/1.31 × 10^−3^	4.84 × 10^−5^/2.40 × 10^−4^
	HHO	3.90 × 10^−12^/8.34 × 10^−12^	8.54 × 10^−16^/7.84 × 10^−16^	7.54 × 10^−11^/1.04 × 10^−10^	1.43 × 10^−11^/1.91 × 10^−11^
	TMCHHO	**2.31 × 10^−14^/4.98 × 10^−13^**	2.05 × 10^−19^/4.74 × 10^−17^	**3.24 × 10^−13^/5.67 × 10^−12^**	**6.53 × 10^−14^/1.23 × 10^−12^**
*f*_3_(*x*)	BOA	1.18 × 10^−5^/1.73 × 10^−5^	7.60 × 10^−6^/1.29 × 10^−5^	1.63 × 10^−5^/2.19 × 10^−5^	2.14 × 10^−6^/2.59 × 10^−6^
	WOA	4.33 × 10^3^/3.18 × 10^5^	5.23 × 10^2^/1.51 × 10^5^	9.68 × 10^3^/5.16 × 10^5^	2.64 × 10^3^/1.00 × 10^5^
	PSO	5.72 × 10^2^/1.09 × 10^5^	2.27/6.79 × 10^04^	5.14 × 10^3^/1.58 × 10^5^	1.24 × 10^3^/2.41 × 10^4^
	DOA	6.75 × 10^−10^/2.32 × 10^−08^	**3.77 × 10^−143^/2.08 × 10^−87^**	1.98 × 10^−8^/6.93 × 10^−7^	3.61 × 10^−9^/1.27 × 10^−7^
	HHO	**9.33 × 10^−20^**/5.91 × 10^−13^	7.69 × 10^−29^/4.03 × 10^−25^	**1.06 × 10^−18^**/1.77 × 10^−11^	**2.64 × 10^−19^**/3.24 × 10^−12^
	TMCHHO	1.36 × 10^−19^/**1.47 × 10^−13^**	1.06 × 10^−31^/7.46 × 10^−28^	1.88 × 10^−18^/**3.93 × 10^−12^**	4.11 × 10^−19^/**7.16 × 10^−13^**
*f*_4_(*x*)	BOA	8.88 × 10^−4^/1.27 × 10^−3^	5.47 × 10^−4^/9.67 × 10^−4^	1.18 × 10^−3^/1.45 × 10^−3^	1.48 × 10^−4^/1.15 × 10^−4^
	WOA	1.72 × 10/7.41 × 10	5.76 × 10^−1^/1.84 × 10^−1^	6.50 × 10/9.54 × 10	1.55 × 10/2.20 × 10
	PSO	4.36/9.21 × 10^01^	5.33 × 10^−1^/8.59 × 10	1.46 × 10/9.62 × 10	3.64/2.76
	DOA	7.68 × 10^−7^/3.15 × 10^−8^	**4.11 × 10^−51^/4.21 × 10^−39^**	1.94 × 10^−5^/5.32 × 10^−7^	3.58 × 10^−6^/1.12 × 10^−7^
	HHO	3.73 × 10^−13^/1.86 × 10^−12^	7.21 × 10^−17^/9.73 × 10^−17^	3.87 × 10^−12^/2.07 × 10^−11^	8.62 × 10^−13^/4.46 × 10^−12^
	TMCHHO	**1.82 × 10^−14^/6.19 × 10^−14^**	3.28 × 10^−18^/1.90 × 10^−18^	**2.40 × 10^−13^/7.70 × 10^−13^**	**5.32 × 10^−14^/1.63 × 10^−13^**
*f*_5_(*x*)	BOA	3.22 × 10/3.46 × 10^−1^	8.39 × 10^−4^/2.10 × 10^−5^	6.10 × 10/4.64	2.21 × 10/1.05
	WOA	8.04/3.03 × 10^−9^	2.84 × 10^−14^/0	6.84 × 10/4.22 × 10^−8^	1.78 × 10/9.51 × 10^−9^
	PSO	2.18 × 10/4.32 × 10^2^	5.02/2.78 × 10^2^	6.26 × 10/6.13 × 10^2^	1.39 × 10/8.12 × 10
	DOA	6.98 × 10^−4^/3.74 × 10^−9^	**0/0**	2.08 × 10^−2^/1.09 × 10^−7^	3.80 × 10^−3^/1.99 × 10^−8^
	HHO	**0/0**	**0/0**	**0/0**	**0/0**
	TMCHHO	**0/0**	**0/0**	**0/0**	**0/0**
*f*_6_(*x*)	BOA	9.00 × 10^−4^/9.49 × 10^−4^	6.85 × 10^−4^/7.96 × 10^−4^	1.20 × 10^−3^/1.18 × 10^−3^	1.08 × 10^−4^/8.73 × 10^−5^
	WOA	3.08 × 10^−7^/5.86 × 10^−7^	1.11 × 10^−9^/8.63 × 10^−9^	1.61 × 10^−6^/7.90 × 10^−6^	4.19 × 10^−7^/1.43 × 10^−6^
	PSO	6.54 × 10^−1^/2.00 × 10	3.61 × 10^−6^/1.86 × 10	2.58/2.07 × 10	8.52 × 10^−1^/5.78 × 10^−1^
	DOA	7.19 × 10^−7^/1.19 × 10^−8^	**8.88 × 10^−16^/8.88 × 10^−16^**	2.05 × 10^−5^/2.53 × 10^−7^	3.74 × 10^−6^/4.67 × 10^−8^
	HHO	8.23 × 10^−13^/1.91 × 10^−12^	4.44 × 10^−15^/4.44 × 10^−15^	1.57 × 10^−11^/3.63 × 10^−11^	2.90 × 10^−12^/6.88 × 10^−12^
	TMCHHO	**1.82 × 10^−14^/1.24 × 10^−14^**	**8.88 × 10^−16^/8.88 × 10^−16^**	**2.71 × 10^−13^/1.04 × 10^−13^**	**5.61 × 10^−14^/2.57 × 10^−14^**
*f*_7_(*x*)	BOA	1.27 × 10^−4^/5.49 × 10^−5^	3.76 × 10^−5^/4.04 × 10^−5^	5.30 × 10^−4^/6.71 × 10^−5^	1.10 × 10^−4^/6.60 × 10^−6^
	WOA	2.21 × 10^−1^/1.20 × 10^−1^	1.11 × 10^−16^/6.88 × 10^−15^	1.04/1.01	2.95 × 10^−1^/3.12 × 10^−1^
	PSO	2.31 × 10^−1^/1.47 × 10^2^	2.49 × 10^−2^/2.70 × 10	8.00 × 10^−1^/3.81 × 10^2^	1.70 × 10^−1^/9.41 × 10
	DOA	4.41 × 10^−10^/5.79 × 10^−8^	**0/0**	7.58 × 10^−9^/1.73 × 10^−6^	1.49 × 10^−9^/3.16 × 10^−7^
	HHO	**0/0**	**0/0**	**0/0**	**0/0**
	TMCHHO	**0/0**	**0/0**	**0/0**	**0/0**

**Table 4 sensors-23-04942-t004:** Statistics of RMS values of 20 experimental results (Note: The best results are highlighted in bold).

Algorithms	Indexes	Armadillo	Bunny	Dragon	Happy
TMCHHO	Ave	**1.85 × 10^−2^**	**1.46 × 10^−2^**	**2.60 × 10^−2^**	**1.17 × 10^−2^**
	Best	1.37 × 10^−3^	9.97 × 10^−4^	4.68 × 10^−3^	1.51 × 10^−3^
	Worst	**4.89 × 10^−2^**	**3.98 × 10^−2^**	**6.16 × 10^−2^**	**3.53 × 10^−2^**
	Std	1.24 × 10^−2^	1.13 × 10^−2^	1.58 × 10^−2^	**8.30 × 10^−3^**
HHO	Ave	3.87 × 10^−2^	5.39 × 10^−2^	5.05 × 10^−2^	5.23 × 10^−2^
	Best	3.06 × 10^−2^	2.21 × 10^−2^	1.79 × 10^−2^	2.18 × 10^−2^
	Worst	5.13 × 10^−2^	6.82 × 10^−2^	6.47 × 10^−2^	8.15 × 10^−2^
	Std	**4.51 × 10^−3^**	**1.05 × 10^−2^**	**1.08 × 10^−2^**	1.70 × 10^−2^
DOA	Ave	3.83 × 10^−2^	4.55 × 10^−2^	4.52 × 10^−2^	3.88 × 10^−2^
	Best	7.44 × 10^−3^	**2.52 × 10^−4^**	1.18 × 10^−5^	**1.32 × 10^−5^**
	Worst	6.17 × 10^−2^	9.69 × 10^−2^	8.76 × 10^−2^	7.15 × 10^−2^
	Std	1.58 × 10^−2^	2.28 × 10^−2^	2.90 × 10^−2^	2.24 × 10^−2^
PSO	Ave	3.48 × 10^−2^	5.33 × 10^−2^	5.36 × 10^−2^	6.01 × 10^−2^
	Best	**1.54 × 10^−7^**	2.44 × 10^−2^	**5.64 × 10^−15^**	1.55 × 10^−2^
	Worst	5.99 × 10^−2^	8.82 × 10^−2^	8.83 × 10^−2^	1.07 × 10^−1^
	Std	1.41 × 10^−2^	2.05 × 10^−2^	2.08 × 10^−2^	1.78 × 10^−2^
WOA	Ave	4.43 × 10^−2^	5.75 × 10^−2^	4.98 × 10^−2^	4.90 × 10^−2^
	Best	3.02 × 10^−2^	2.17 × 10^−2^	3.44 × 10^−3^	3.86 × 10^−3^
	Worst	6.12 × 10^−2^	9.21 × 10^−2^	8.48 × 10^−2^	9.87 × 10^−2^
	Std	9.02 × 10^−3^	2.04 × 10^−2^	2.81 × 10^−2^	2.85 × 10^−2^
BOA	Ave	6.09 × 10^−2^	7.56 × 10^−2^	1.02 × 10^−1^	9.70 × 10^−2^
	Best	3.94 × 10^−2^	5.91 × 10^−2^	6.15 × 10^−2^	4.10 × 10^−2^
	Worst	9.55 × 10^−2^	1.16 × 10^−1^	2.48 × 10^−1^	1.61 × 10^−1^
	Std	1.54 × 10^−2^	1.44 × 10^−2^	5.32 × 10^−2^	2.59 × 10^−2^

**Table 5 sensors-23-04942-t005:** The statistical data of final registration results.

Method	Armadillo	Bunny	Dragon	Happy
ICP	1.49 × 10^−02^	1.75 × 10^−02^	9.82 × 10^−03^	2.99 × 10^−02^
TMCHHO + ICP	1.34 × 10^−06^	4.33 × 10^−04^	6.45 × 10^−04^	5.52 × 10^−04^

**Table 6 sensors-23-04942-t006:** The size of the point cloud dataset.

Survey Area	Number of Preprocessing Points (Local)	Number of Postprocessing Points (Local)
3D laser scanning	303,919	68,905
UAV tilt photography	503,555	62,300

**Table 7 sensors-23-04942-t007:** The statistical data analysis of fusion accuracy.

Survey Area	Avg	Best	Worst	Std
①	0.0311	0	0.3603	0.0644
②	0.0182	0	0.5154	0.0551
③	0.0190	0	0.5196	0.0550
④	0.0186	0	0.2361	0.0525
⑤	0.0304	0	0.3881	0.0666

## Data Availability

Some data used of this study are available from the corresponding author, upon reasonable request.

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
