# Peer review of "A Cross-Source Point Cloud Registration Algorithm Based on Trigonometric Mutation Chaotic Harris Hawk Optimisation for Rockfill Dam Construction"

_sensors, 2023, doi:10.3390/s23104942_

Round 1

Reviewer 1 Report

The article accomplishes point cloud alignment through two improvements:

1. Enhancing data collection through an airborne integrated environmental information sensing strategy with multiple mapping systems.

2. Complete spatial coordinate unification of multimodal data by improved TMCHHO

The article also does a good job of completing tests on different data sets, but there are still a few shortcomings:

1. The article ablation experiment results data is not shown, and the ablation experiment only Figure 13, not very obvious.

2. The final alignment algorithm results show less, suggest more results show.

3. The algorithm is not compared with the newer alignment algorithm, and it is suggested to attach the experimental diagram for comparison with the latest algorithm.

The writing of the paper needs to be polished.

Reviewer 2 Report

The article introduces a cross-source point cloud registration method that combines the trigonometric mutation chaotic Harris hawks optimization (TMCHHO) coarse registration algorithm and the iterative closest point (ICP) fine registration algorithm. Compared with the realistic modeling solutions of a single mapping system, the accuracy and integrity of the fusion model have been improved. The article has innovation, but there are certain deficiencies in writing.

1、  Page3. Line105., please add references to PWLCM to help readers understand it better.

2、  Page5. Line177., the abbreviation PWLCM stands for "the piecewise linear chaotic map" as noted in the original text. Adding "chaotic mapping" after it may be redundant.

3、  Page4. Figure1., please unify the titles of the pictures at the bottom or top.

4、  Page5. Equation1., equation 1 describes the rigid transformation of 3D point clouds. We are more interested in your innovative content, so you do not need to repeat this description. "The explanation for the ICP algorithm and HHO method in the following text is similar. We will also focus more on the complete process of the TWCHHO algorithm."

5、  Page10. Equation13., what does 'K' represent? The original text should add a complete annotation for readers' understanding.

6、  Page22. Figure13., what does the blue part in the image represent? If it is point cloud data, readers may have difficulty obtaining information other than color, such as texture and shape, making it difficult for them to understand whether the registration process is accurate in terms of visualization.

7、  Page24. Figure16., the image displaying the registration result is abstract, and only from a top-down view, it is indeed difficult to discern the accuracy of the registration structure.

Reviewer 3 Report

Dear authors, thank you for an interresting contribution.

It is focused on creating and processing point clouds, based on photogrammetric measurement using drone and TLS (terrestrial laser scanning).

I have only small remarks:

1) UAV - it is a very frequently used abbreviation. But it defines (not very correctly) an unmanned aerial vehicle. If so, it would be better to use UAS (unmanned aerial system) - it includes the ground segment, i.e. the pilot and the control equipment. This is a question for discussion. The correct abbreviation is probably RPAS (remotely piloted aerial system).

2)If you are talking about combining different geodetic data, you need to have some more precise measurements to verify the result. So, for example, geodetic measurements and the use of control points.

3)You are verifying functionality on small models ( bunny, etc.) and extrapolating to a dam; this is not entirely correct.

4) you perform a classic geodetic activity - merging data from multiple technologies. For this reason add more citations and some sentences on geodetical measurement technologies in introduction on geodetic measurements; this is typical - joining of different data into one model. You can find a lot of articles on this theme; most use verification of accuracy by geodetic methods. This should be mentioned. But I understand your goal was data reduction. Still, add it.

joining of two models, for example

/10.3390/rs15010131

or

10.3390/app11020754

or

10.2312/VAST/VAST06/163-170

May be, add a benefit of your solution in conclusion (saving data, time, better model, etc.). this will improve the readability of the article for other users

Reviewer 4 Report

The authors define their objectives clearly and also point out the limitations of their work.

I appreciate it as well as the way they present their work.

The structure and style of the paper is adequate. Each section has a clear role, just like the figues and tables have.

Description of the framework and the different computations is clear, and the result of the work can be considered as an added value to this field.
